# High-resolution definition of humoral immune response correlates of effective immunity against HIV

Galit Alter[1], Karen G Dowell[2], Eric P Brown[3], Todd J Suscovich[1], Anastassia Mikhailova[1], Alison E Mahan[1], Bruce D Walker[1], Falk Nimmerjahn[4], Chris Bailey-Kellogg[2] & Margaret E Ackerman[3,*]

## Abstract

**Defining correlates of immunity by comprehensively interrogating the extensive biological diversity in naturally or experimentally protected subjects may provide insights critical for guiding the development of effective vaccines and antibody-based therapies. We report advances in a humoral immunoprofiling approach and its application to elucidate hallmarks of effective HIV-1 viral control. Systematic serological analysis for a cohort of HIV-infected subjects with varying viral control was conducted using both a high-resolution, high-throughput biophysical antibody profiling approach, providing unbiased dissection of the humoral response, along with functional antibody assays, characterizing antibody-directed effector functions such as complement fixation and phagocytosis that are central to protective immunity. Profiles of subjects with varying viral control were computationally analyzed and modeled in order to deconvolute relationships among IgG Fab properties, Fc characteristics, and effector functions and to identify humoral correlates of potent antiviral antibody-directed effector activity and effective viral suppression. The resulting models reveal multifaceted and coordinated contributions of polyclonal antibodies to diverse antiviral responses, and suggest key biophysical features predictive of viral control.**

**Keywords** antibody; effector function; HIV; systems serology
**Subject Categories** Genome-Scale & Integrative Biology; Immunology; Systems Medicine
**Mol Syst Biol. (2018) 14: e7881**

## Introduction

Vaccines have been a remarkable public health success, with over 60 licensed vaccines estimated to have saved millions of lives. Yet, effective vaccines are still urgently needed for a number of devastating infectious agents. For many of these pathogens, conventional empirical approaches have failed to yield broad protection, and traditional measures of immunity, including antibody titers and *in vitro* neutralization, have proved insufficient to predict protection. Promisingly, the empirical processes of vaccine design used over the past 200 years are being radically revised to leverage crucial insights into correlates of immunity revealed by studies of naturally or experimentally protected subjects using an emerging suite of systems vaccinology approaches (Pizza *et al*, 2000; Pulendran, 2009; Rappuoli & Aderem, 2011; De Gregorio & Rappuoli, 2012; Sekaly & Pulendran, 2012). However, in contrast to transcriptional and cellular profiling tools that have yielded substantial and detailed insights into the mechanisms underlying the induction of more effective humoral immune responses (Querec *et al*, 2009; Nakaya *et al*, 2011; Lin *et al*, 2015; Kazmin *et al*, 2017), the current tool kit for the evaluation of vaccine-induced humoral responses remains insufficient.

Whereas there is accumulating evidence that antibody effector functions, including complement fixation and phagocytosis, often play a central role in protective immunity, functional antibody assays and broad immunoprofiling approaches have been challenging to implement widely or consistently in research and development platforms. As a result, only limited insights into correlates of potent humoral responses can generally be made. Approaches that may close the gap on our ability to better define humoral correlates of immune protection could accelerate the development of more effective therapeutics and vaccines. Thus, high-resolution and high-throughput experimental and analytical means to profile and dissect the extensive natural biodiversity in the humoral immune response induced by natural infection or following vaccination have the potential to support efficient identification of key correlates of immunity useful for both future vaccine evaluation as well as in the design of monoclonal antibody therapeutics. Here, we aimed to merge comprehensive functional and biophysical antibody assessments with computational analyses to develop tools to define the minimal biomarkers that track with desirable clinical outcomes.

While several lines of evidence point to a role for cytolytic antibodies in viral control of HIV (reviewed in Lewis, 2014), little is known about the specific biophysical features of the antibodies that drive cytolytic activity and control virus replication most effectively. Promisingly, a minor subset of infected subjects are known to control HIV in the absence of anti-retroviral therapy, such as HIV-1

1 Ragon Institute of MGH, MIT and Harvard, Cambridge, MA, USA
2 Department of Computer Science, Dartmouth College, Hanover, NH, USA
3 Thayer School of Engineering, Dartmouth College, Hanover, NH, USA
4 Department of Biology, Institute of Genetics, University of Erlangen-Nuremberg, Erlangen, Germany
*Corresponding author. Tel: +1 603 646 9922; E-mail: margaret.e.ackerman@dartmouth.edu

controllers include elite controllers (EC) with undetectable viremia, and viremic controllers (VC) who suppress the virus to nearly non-transmissible levels. While unique cytotoxic T-cell responses are enriched in a fraction of controllers (Fellay *et al*, 2007; Kosmrlj *et al*, 2010), previous studies have highlighted enhanced antibody effector function and altered subclass distribution among both long-term non-progressive subjects (Lal *et al*, 1991; Baum *et al*, 1996; Forthal *et al*, 1999; Banerjee *et al*, 2010; French *et al*, 2013; Ackerman *et al*, 2016) and protective vaccine recipients (Haynes *et al*, 2012; Tomaras *et al*, 2013; Chung *et al*, 2014; Yates *et al*, 2014). Likewise, differences in the antigen specificity of the humoral response have also been noted in these populations (Lal *et al*, 1991; Hogervorst *et al*, 1995; Banerjee *et al*, 2010; Haynes *et al*, 2012; French *et al*, 2013), suggesting the value of comprehensive IgG Fab, Fc, and functional profiling to inform the identification of humoral mechanisms of action and associative relationships to viral suppression.

Here, systematic antibody profiling of an unprecedented array of antibody features and effector functions was conducted across a cohort of HIV-infected subjects with varying viral control and disease progression to deconvolute and dissect relationships among IgG Fab-mediated antigen recognition, Fc-mediated innate immune receptor binding characteristics, and effector functions. These rich serological data were modeled with the objective to define the specific polyclonal humoral correlates of effective viral suppression and of potent antiviral antibody effector activity. The resulting models point to the multifaceted and coordinated contributions of antibodies to diverse antiviral responses, and the minimal biophysical features that predict virus control. By modeling their interactions to define the underlying principles by which antibodies collaborate and/or compete to dictate the overall antiviral activity of the humoral response, we define new biomarkers associated with antiviral antibody effector function, aviremia, and non-progression.

# Results

A systems serology approach was used to comprehensively profile antibody features and functions spanning from antigen recognition through effector cell activation (Fig 1). We evaluated antibody specificity across 41 different HIV protein variants, and innate immune recruiting capacity by determining the titer, subclass, glycosylation, and FcγR and lectin recognition properties of these antigen-specific antibodies (Appendix Table S1, Dataset EV1). These methods were coupled to functional assays including assessments of antibody-dependent cellular cytotoxicity (ADCC), complement deposition (ADCD), neutrophil phagocytosis (ADNP), and natural killer (NK) cell activation (NKA) as measured by CD107a, IFNγ, and MIP1β expression—antiviral activities that have correlated with protection from infection in animal models (Barouch *et al*, 2013, 2015; Fouts *et al*, 2015; Bradley *et al*, 2017). Data were collected for a blinded cohort of 200 HIV-infected subjects that included EC, VC, and chronic progressive subjects on (treated progressor, TP) or off (untreated progressor, UP) anti-retroviral therapy.

Given the rich data collected, a machine-learning approach was employed to identify combinations of humoral response features able to predict subject class and antibody effector function. Cross-validated classifiers trained to distinguish subject groups identified minimal sets of antibody features that accurately and robustly discriminated among all four subject groups, between viremic and aviremic subjects, and between controllers and progressors (Fig 2). Subject class was determined by a class score (LOD, or $\log_2$ odds ratio) that defined the relative likelihood of a given subject's assignment to one as compared to other classes. Differentiation across EC, VC, TP, and UP groups was accomplished with approximately 60% accuracy, as compared to the approximately 25% accuracy expected at random based on class size for this four-group differentiation

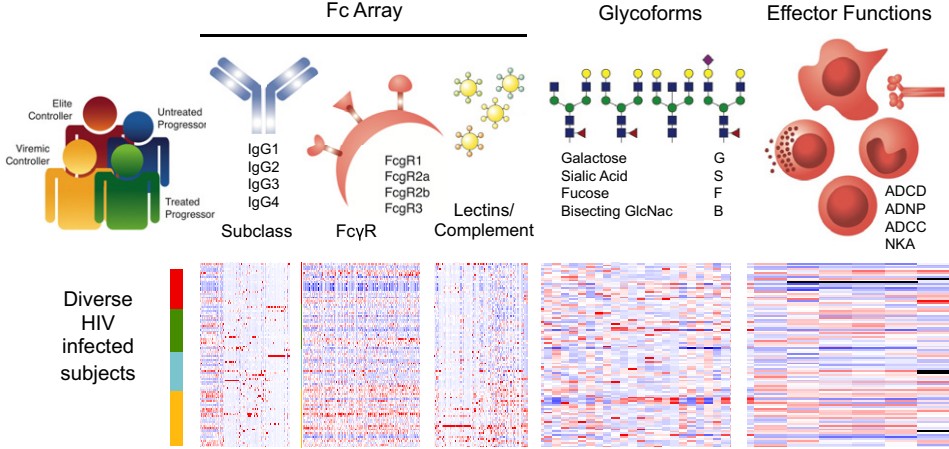

**Figure 1. Serological profiling.**

Systematic immune profiling was conducted to characterize humoral responses observed in HIV-infected subjects including elite controllers (EC), treated progressors (TP), untreated progressors (UP), and viremic controllers (VC). For a set of 41 different virally encoded proteins comprised primarily of variants of envelope (gp140, gp120, gp41), capsid (p24), and others (e.g., integrase, Nef, Vif, Rev), the Fc array was used to determine the subclass (IgG1, 2, 3, 4), ligation propensity for Fc receptor (FcγR1, 2a, 2b, 3a, 3b) and complement cascade initiating proteins (C1q and MBL), and glycosylation state (plant-derived lectins) for antibodies specific to each tested antigen. Antibody glycoforms were assessed for total serum IgG and for gp120-specific antibodies using capillary electrophoresis to define specific glycoforms and overall levels of antibody galactosylation (G), sialylation (S), fucosylation (F), and GlcNac bisection (B). Antibody effector functions including antibody-dependent complement deposition (ADCD), antibody-dependent neutrophil phagocytosis (ADNP), antibody-dependent cellular cytotoxicity (ADCC), and NK cell activation (NKA), defined by expression of CD107a, MIP1β, and IFNγ were assessed using high-throughput functional assays.

model, and as observed when scrambled study data were used as a modeling input (Fig 2A). Approximately 75% accuracy was observed for the two-way classifications aimed at differentiation of subjects by progression (EC and VC versus TP and UP) or by viremia (EC and TP versus VC and UP), as compared to the approximately 50% accuracy expected by chance, or again, observed when study data were permuted prior to learning (Fig 2A). Confusion matrices (Fig 2B–D), which compare actual and predicted classifications, indicated that with the exception of UPs, who could not be confidently differentiated from TPs or VCs, most classes were predicted well. While there was little evidence of systematic confusion, among misclassified VCs and TPs, most were modeled to be UPs, consistent with the difficulty noted in classification of this group, and suggestive that they may have a less distinct humoral profile as a group. Further, class score can be considered a measure of model confidence (Fig 2B–D). Whereas in the two-way

classifications misclassified subjects often had marginal class scores, in the four-group differentiation models, incorrect predictions were often made confidently. For example, misclassified ECs tended to be confidently predicted as either TPs or VCs, suggestive of the existence of some subjects with profiles that are considerably more consistent with other classes.

Again, in comparison, models trained on permuted, that is, intentionally scrambled, data failed to provide classification accuracy beyond that expected at random based on class size (Fig 2A). Given this indicator of model reliability, inspection of the specific antibody features enabling robust classification has the potential to suggest meaningful and mechanistic associations with viremia and progression (Appendix Figs S1–S3). Consistent with prior studies, differential aspects of antibody glycosylation (Ackerman *et al*, 2013a), FcγR ligation (Ackerman *et al*, 2013b), and antigen specificity and subclass (Lal *et al*, 1991; Banerjee *et al*, 2010; French *et al*, 2013;

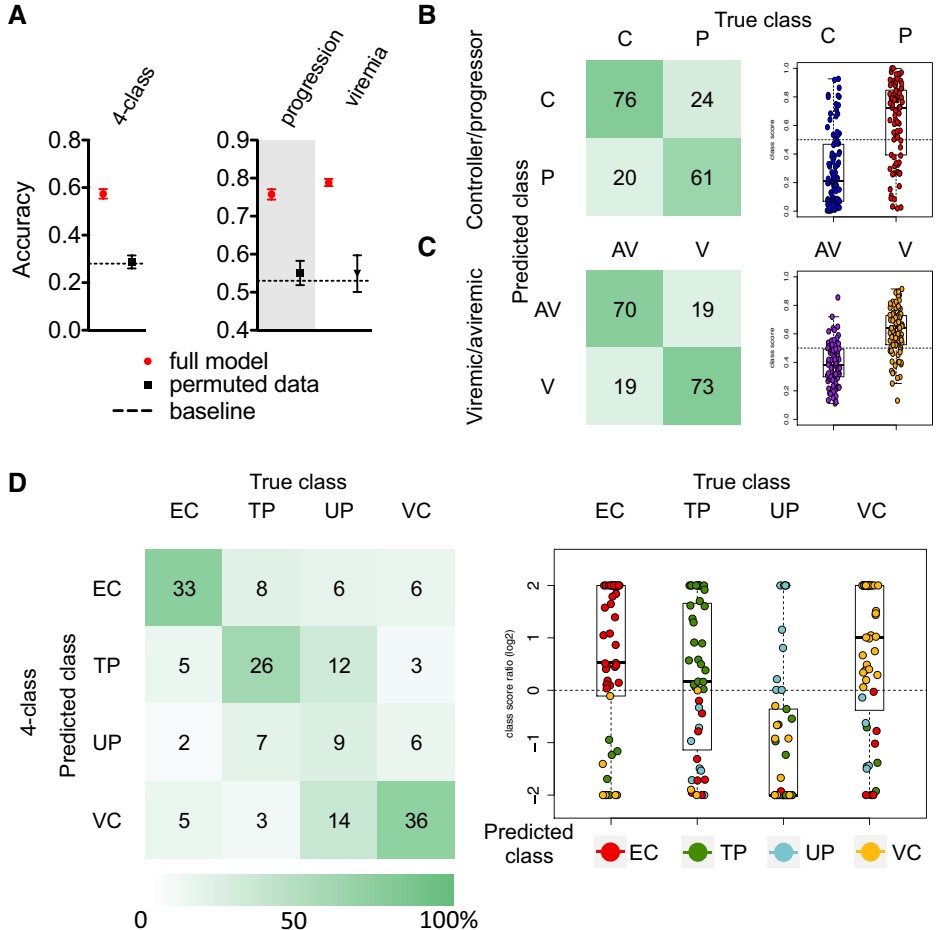

**Figure 2. Humoral profiles distinguish subject class.**

A Accuracy of classification models trained using antibody profiles to distinguish across all four infected subject classes (EC, elite controller; TP, treated progressor; UP, untreated progressor; VC, viremic controller) or between subjects with variable progression [controllers (C) defined as EC and VC versus progressors (P) defined as TP and UP] or variable viremia [viremics (V) defined as UP and VC versus aviremics (AV) defined as EC and TP]. Accuracies observed in models (red) are compared to the baseline expectation based on random chance (dotted line) or when classification models were constructed using permuted class labels (black). Error bars denote standard deviation observed across 100 cross-validation folds and replicates.

B–D Confusion matrices (left) and class log₂ odds scores (right) for prediction of progression (B), viremia (C), and subject class (D). In the four-class model, the class assigned to each subject is indicated by color (EC red; TP green; UP blue; VC yellow). For (D), the plot was truncated to give |class score| ≤ 2. Box and whisker plots denote median, minimum, maximum, and interquartile ranges.

Ackerman *et al*, 2016) each contributed to successful class differentiation (Dataset EV2).

This modeling approach reliably identified Fab and Fc characteristics that were distinct among subject groups using traditional statistical approaches. For example, while HIV infection is known to alter the global glycosylation state of plasma IgG (Moore *et al*, 2005) in both the presence and absence of plasma viremia (Ackerman *et al*, 2013a), this dataset offered an unprecedented opportunity to identify IgG glycoforms that might be associated with variable viral control or progressive infection. For example, numerous bisected glycoforms contributed to the differentiation of aviremic from viremic subjects when the glycosylation profiles of HIV envelope-specific antibodies were used to predict viral load status (Appendix Fig S3), suggesting a difference in the level of antibody bisection between groups. Indeed, aviremic subjects were distinguished by an enrichment of envelope-specific antibodies with bisected glycans (Fig 3A, Appendix Fig S4). Interestingly, this distinction was a specific attribute of pathogen-directed antibodies rather than a general feature of plasma antibodies in aviremic subjects, as it was restricted to the envelope-specific antibody pool, and somewhat reduced bisection was observed on total plasma IgG in aviremic subjects relative to those with ongoing viral replication. Whether this glycosylation state difference associated with viral load may be a cause or effect of viremic status remains to be determined, but potential mechanistic relevance to antiviral humoral immunity is suggested by previous studies demonstrating that antibodies with bisected glycans exhibit enhanced FcγR3 binding and ADCC activity (Umana *et al*, 1999), a mechanistic link that was further explored within this dataset in models of antibody function.

Analysis of the Fab specificity of features contributing to classification of ECs but not observed in modeling viremia status (Appendix Figs S1 and S2, and Dataset EV2) pointed toward recognition of the virus capsid (p24/Gag) as a differential marker between subjects with immune-mediated as opposed to pharmacologic viral suppression. To investigate this antigenic target as compared to other biomarkers that could distinguish ECs and TPs, the two classes of aviremic subjects were plotted to identify disparities in the humoral response that may be associated with the mechanism of viral suppression (Fig 3B). As indicated by a volcano plot of Fc array measurements, ECs had distinctly elevated levels of capsid-specific antibodies, consistent with previous observations in persons with non-progressive infection (Hogervorst *et al*, 1995). HIV-specific IgG2 and IgG4 response magnitudes were greater in TPs, particularly among antibodies to gp41, whereas FcγR-ligating antibodies to internal HIV proteins beyond the capsid were also enriched among ECs, perhaps as a result of variation in antigen processing, presentation, and T-cell activation in this subject group.

Thus, the profiling approach used here can serve to confirm previously observed and identify novel aspects of the humoral response to HIV infection that differentiate among subject groups. Significantly, features of both antibody specificity and Fc characteristics revealed critical compound distinctions in the humoral responses of subjects associated with varying viral suppression, progression, and treatment status.

Because mounting evidence points to coordination of multiple aspects of humoral responses as superior markers of both reduced risk of infection and durable viral control (Chung *et al*, 2014; Barouch *et al*, 2015; Ackerman *et al*, 2016), we also evaluated correlative relationships between antibody glycoforms and biophysical antibody features for each subject group (Fig 4). Balancing the competing desires of discovery and confidence, plots depicting relationships between glycoforms and Fc Array measurements with correlation coefficients exceeding an absolute value magnitude of 0.4 and an uncorrected *P*-value of 0.01, representing the top 7% of correlations in strength and significance, were generated. Striking differences were apparent in the coordination of the response among groups, evidenced by distinct hub and spoke linkages. Among ECs, who possess preserved polyfunctional responses (Ackerman *et al*, 2016), the IgG G1S1F glycoform was uniquely correlated with the induction of multiple IgG3 specificities. In contrast, the G2S2F

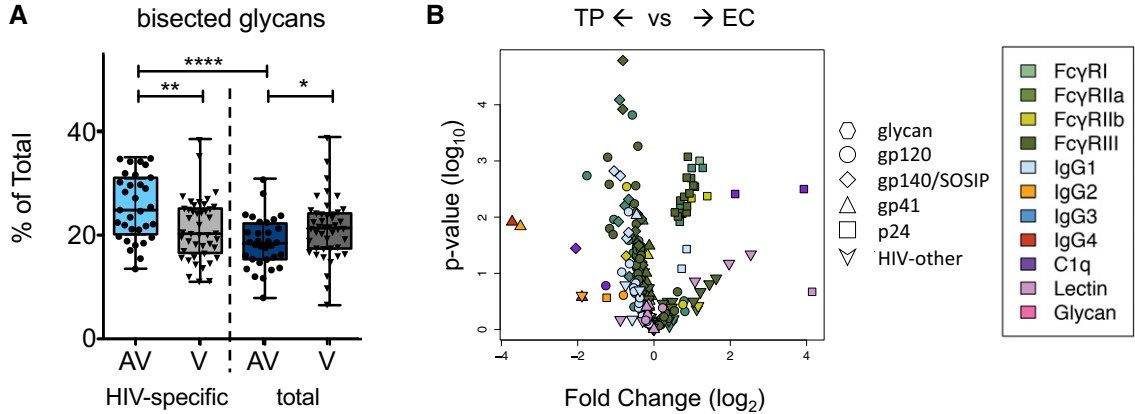

**Figure 3.   Examples of differentiating humoral response features identified by classification models.**

A   Level of IgG glycan bisection among HIV-specific and total plasma IgG in viremic (V) and aviremic (AV) subjects. Boxes and whiskers indicate interquartile range, median, minimum, and maximum values. Significance in differences between groups as defined by an uncorrected two-tailed unpaired *t*-test is indicated as *$P$ < 0.05, **$P$ < 0.005, ****$P$ < 0.0001.

B   Volcano plot characterizing the magnitude (fold change) and significance (*P*-value) of differences in Fc array measurements between the two types of aviremic subjects, elite controllers (EC) versus treated progressors (TP). Fc array measurements are colored by detection reagent and antigen specificity is indicated by symbol shape.

glycoform was uniquely and strongly associated with IgG2 responses among UP. Because individual antibody types act in the context of immune complexes, such variance in the architecture of humoral responses may identify global differences among responders and between responses, informing our understanding of

antibody phenotype networks and both mechanisms and biomarkers of enhanced viral control.

We next sought to define the composite features of the humoral immune response predictive of the potency of antibody effector activity, employing biophysical data to develop cross-validated

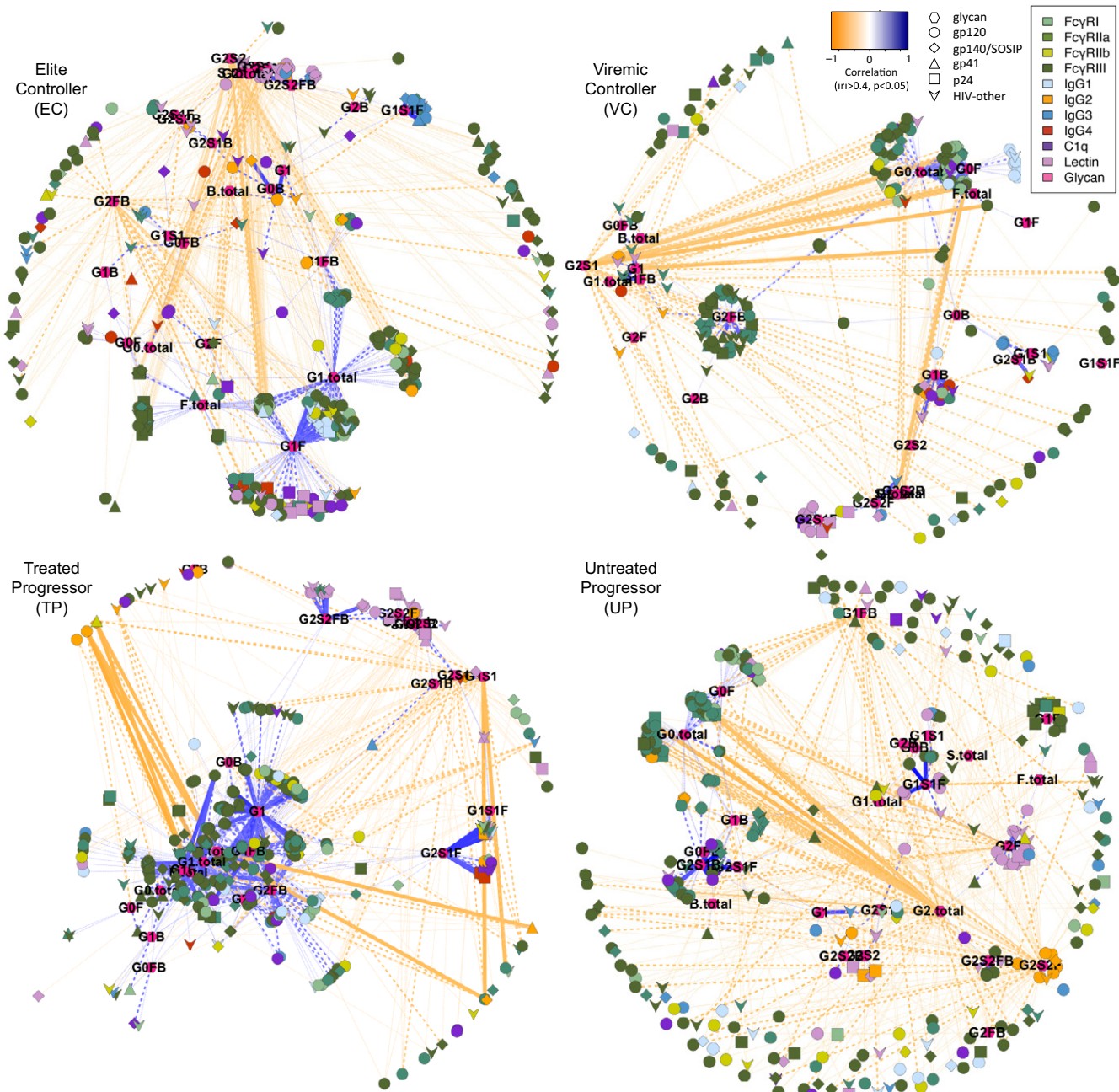

**Figure 4.  Key distinctions in humoral immune responses among subject groups.**
Network plots of significant correlations (|r| > 0.4, P < 0.05) between IgG glycoforms and Fc array data within each subject group. Glycans are labeled and represented by pink hexagons, with strength and direction of correlation to Fc array measurements illustrated in distance and color (positive correlations are proximal and colored in blue, negative correlations are distal and noted in orange). Line weight and style indicate significance in uncorrected two-tailed correlation tests, where thin dotted lines represent P < 0.05, medium dashed lines represent P < 0.01, and thick solid lines represent P < 0.001. Fc array measurements are colored by detection reagent and antigen specificity is indicated by symbol shape.

models of each effector function (Fig 5A). Prediction accuracy, the correlation between modeled and observed activity, was generally as good as the degree of correlation observed between assay replicates, which provides a reasonable benchmark for how well a model might be expected to perform. In contrast, when the same modeling strategy was applied to permuted data, random performance was observed, again, providing confidence that the discovered relationships are reliable. In all cases other than ADCD, in which equivalent performance was observed, models trained on the complete biophysical dataset were more accurate than models that considered only traditional measures of antibody titer, clearly demonstrating that qualitative evaluation of antigen-specific antibodies was important to accurate prediction of functional potency. Despite the heterogeneity among subject groups, we found that for each function, a single model could effectively predict antibody activity equally well across as within groups (Fig 5B, Appendix Fig S5), indicating that relationships between antibody features and antibody functions were common across subjects, and highlighting the strength and potential general utility of these biomarkers in predicting enhanced antibody functionality.

This consistency suggested that the model features may bear mechanistic significance. Qualitative assessments of the FcγR binding capacity of envelope-specific antibodies were the most frequent and strongest predictors of the cell-based functional assays (Fig 6A–D, Appendix Fig S6). Specifically, the ability of envelope-specific antibodies to complex FcγR3 was the greatest and most frequent contributing features to models of effector activity in NK assays. That this effector type expresses only FcγR3 suggests that this profiling approach has captured subtle differences in antibody responses that are associated with differential recruitment among homologous FcγRs. Further, consistent with expectations, C1q recognition was the most important feature in models of complement deposition. Subclass assessments also contributed to ADCD models, with IgG3

responses contributing to enhanced bioactivity. Contrastingly, negative contributions were observed for IgG2 and IgG4, suggesting that these subclasses, which bind poorly to C1q, rather than benignly co-existing, may actively disrupt and thereby reduce initiation of the complement cascade. Evidence for interference between antibody types has been previously suggested by both traditional correlate and machine-learning analyses (Haynes *et al*, 2012; Choi *et al*, 2015; Ackerman *et al*, 2016) and experimentally supported by depletion (Chung *et al*, 2014) and competition (Tomaras *et al*, 2013) experiments. Again, such observations point toward the value of high-resolution assessment of multiple aspects of the humoral response, as well as the exploration of multi-parameter modeling approaches. Thus, even though titer-alone models performed similarly well to those trained with qualitative data for ADCD, mechanistically relevant insights about immune complex-driven complement deposition, with a biological basis supported by previous studies of monoclonal antibodies in other disease settings, were gained from the model features. In contrast, the relatively poorer predictive performance for ADNP suggests that additional features of antigen-specific antibodies, yet to be captured by this platform, may be essential to better model and dissect qualitative differences in induction of effective neutrophil responses.

Lastly, beyond even IgG subclass and FcγR binding, it is well appreciated that antibody glycosylation can dramatically impact antibody bioactivity. For example, Fc fucosylation reduces binding to FcγR3a, thereby regulating ADCC potency (Shields *et al*, 2002), whereas Fc bisection is associated with enhanced FcγR3a binding (Umana *et al*, 1999). Here, while we observed that IgG fucosylation was strongly associated with neither ADCC (Figs 6E, Appendix Fig S7) nor viremia (Appendix Fig S4) in this cohort, beyond differentiating subjects on the basis of viremia, bisection of the IgG glycan was significantly associated with ADCC activity, as were two specific, albeit rare, afucosylated glycoforms (G0B and G1). Other

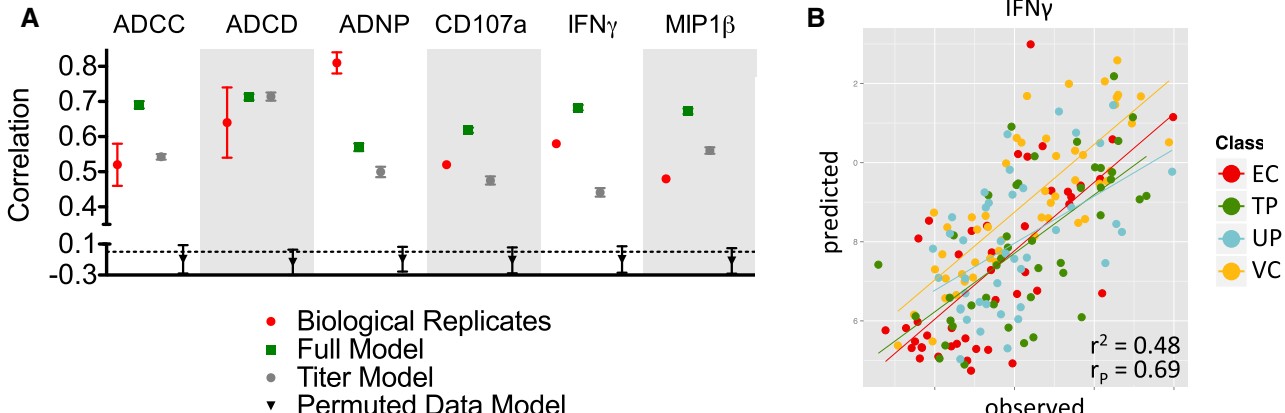

**Figure 5. Diverse antibody activities are robustly predicted by biophysical profiling data.**

A Prediction quality (correlation coefficients) of models learned from Fc array data for each antibody function and as compared to correlations observed between biological assay replicates (red). Agreement between model predictions and experimental observations using complete Fc array data (green), only IgG subclassing data from the Fc array (gray), and when activity values were permuted (black) are shown. The dotted line represents a baseline drawn at a correlation coefficient of 0, the performance expected at random. Symbols and error bars represent the mean and standard deviation across experimental replicates (red, triplicates available only for ADCC, ADCD, and ADNP) or across 100 cross-validation folds and replicates for models (green, gray, black).

B Representative scatterplot of predicted versus observed antibody-dependent expression of IFNγ by NK cells for each study subject. Subjects are color coded by class and the best fit line for each class is illustrated (EC red; TP green; UP blue; VC yellow).

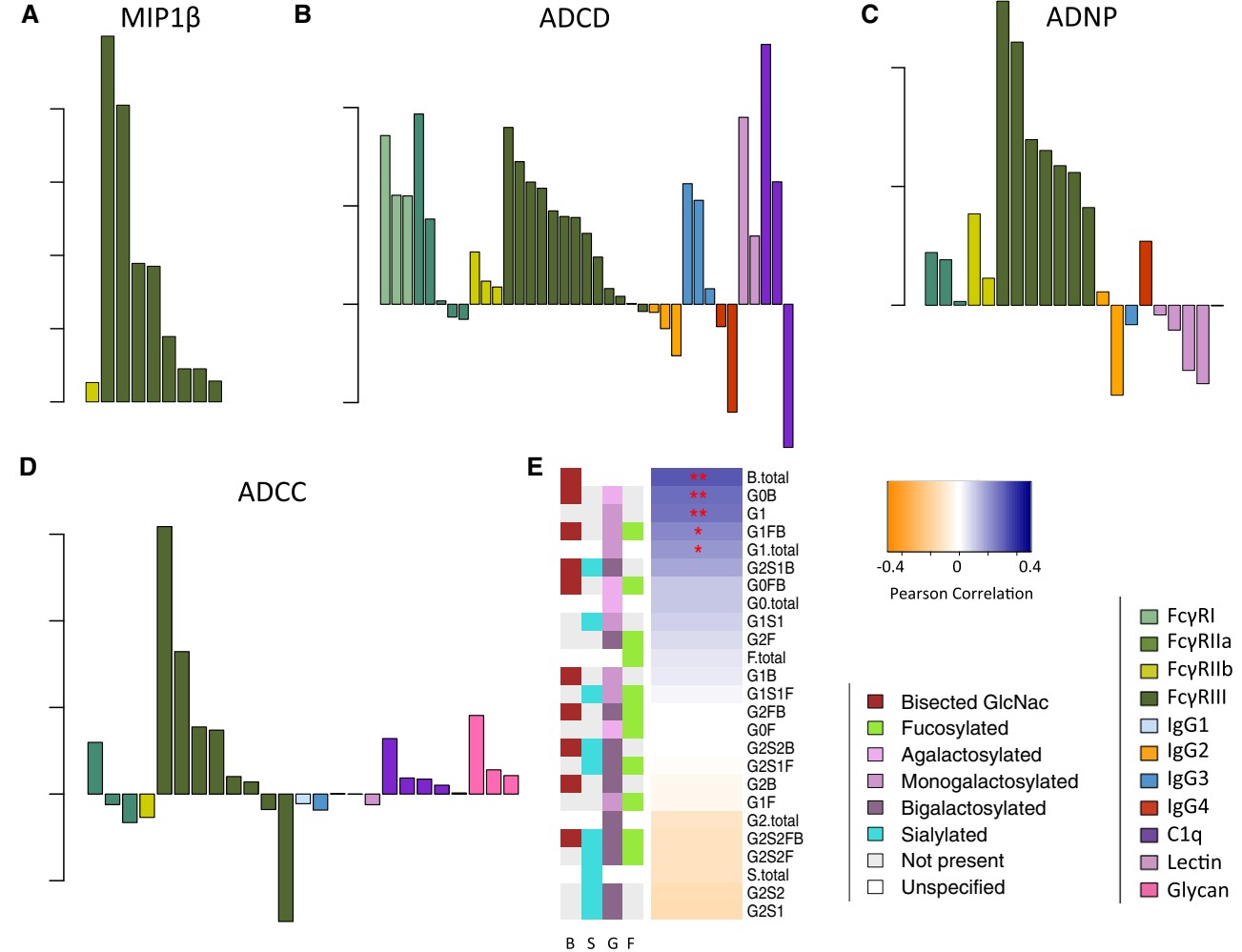

**Figure 6.  Mechanistic insights into effector function afforded by robust predictive models.**

A–C    Relative coefficient weights and identities of Fc Array features contributing to activity models for NK cell MIP1β expression (A, MIP1β), antibody-dependent complement deposition (B, ADCD), and antibody-dependent neutrophil phagocytosis (C, ADNP). Fc array measurements are colored by detection reagent.

D    Relative coefficient weights and identities of Fc Array and gp120-specific antibody glycoform features contributing to ADCC predictions. Fc array measurements are colored by detection reagent and glycan assessments are illustrated in pink.

E    Correlative relationships observed between ADCC and glycoform prevalences. Glycoforms are listed in order of and colored by strength and direction of correlation coefficients (positive correlations in blue, negative correlations in orange). Significance as defined by an uncorrected two-tailed correlation test is indicated as *$P < 0.05$, **$P < 0.01$. Antibody glycan species are annotated according to the presence or absence of GlcNac bisection (B), sialylation (S), galactosylation (G), and fucosylation (F).

activities were more strongly correlated with different glycoforms (Appendix Fig S7), highlighting the extent to which novel glycoforms may be naturally selected to drive enhanced antibody bioactivity. Thus, deeper analysis of natural functional humoral specification may be encoded within subclass and glycoform types to adaptively direct differential effector functions.

# Discussion

This work builds upon the systems serology framework recently used in analysis of human responses to tuberculosis infection and HIV vaccination (Chung *et al*, 2015; Lu *et al*, 2016), in an expanded form that has supported the identification of correlates of protection in nonhuman primate SIV/SHIV challenge studies (Barouch *et al*, 2015; Vaccari *et al*, 2016; Bradley *et al*, 2017), and applies this enhanced systematic humoral profiling approach to capture the spectrum of antigen and innate immune recognition properties of HIV-specific antibodies induced by natural infection. Immune markers of protection represent critical biomarkers that are selectively enriched in individuals that control or resist infection that may guide the development of diagnostics, influence the development of rationally designed vaccines, provide design clues for the generation of more effective monoclonal therapeutics, or simply support the evaluation and down-selection of different vaccine products. However, the discovery of humoral biomarkers, which have the added advantage of being antigen-specific, has historically been narrowed by the limited array of assays that probe the humoral

immune response. Specifically, traditional humoral correlates analyses have focused on the comparison of the magnitude of the humoral immune response to pathogen targets and in some cases to their ability to block infection (neutralization). Yet, antibodies have the capacity to contribute to pathogen control via a diverse array of functions driven by their ability to recruit the innate immune system. However, while novel platform technologies, such as the recently described systems serology approach (Ackerman *et al*, 2017), can provide a comprehensive glimpse at the broad view of the landscape of functional responses of pathogen-specific antibodies, approaches to specifically define minimal physical biomarkers, which can be tailored to maximize information and reduce experimental redundancy, which are more easily interrogated in validated assays, and which are associated with desirable clinical outcomes, need to be further implemented to better define cross-cohort, cross-regimen, and cross-pathogen principles of protective humoral immunity. To this end, the combination of tools that broadly probe antigen specificity, Fc receptor binding, functional activity, and antibody Fc glycosylation as reported here offers a distinct opportunity to chart the integrated Fab and Fc characteristics that are uniquely enriched in individuals that are protected from disease.

Already, for several vaccines, including the meningicoccal B (Goldschneider *et al*, 1969) or Haemophilus influenza (Kayhty *et al*, 1983) vaccines, functional antibody correlates are key to predicting protective immunity (Plotkin, 2010). However, functional antibody assays are more variable and difficult to validate. The humoral immune response to infection is driven by polyclonal pools of antibodies that target the pathogens through swarms of antibodies that interact with both low-affinity classical and non-classical Fc receptors expressed in different combinations on all innate immune cells. Combinatorial binding profiles then drive unique antibody effector functions (Tomaras *et al*, 2013; Chung *et al*, 2014; Pollara *et al*, 2014; Choi *et al*, 2015; Ackerman *et al*, 2016). Thus, biophysical interactions of antigen-specific antibodies with specific Fc receptors, rather than the total levels of pathogen-specific antibodies, are likely to provide enhanced resolution of functional antibodies that track with protective immunity. Biophysical assays, able to provide simple measures that go beyond antibody titer, may provide high value for vaccine evaluation. Importantly, common Fc receptor binding profiles were able to predict antibody functionality across the four different groups of HIV-infected subjects, strongly arguing that the relationships between biophysical antibody characteristics and antibody activities are universally preserved across the groups. Thus, it is possible to identify minimal biophysical features that track with the functions that are selectively enriched in protected populations. While antibody Fc modifications that increase ADCC have been previously defined (Umana *et al*, 1999; Shields *et al*, 2002), the Fc modifications that enable selective augmentation of ADCP and ADCD are less well characterized. Moreover, novel modifications that drive additional functional responses, including the ability of antibodies to induce dendritic cell (DC) activation or anti-inflammatory activity, may be identified and point to additional opportunities to develop more effective therapeutics. Thus, while observations using this approach are necessarily only associative, an integrated Fc-functional and deep biophysical profiling offers a unique opportunity for the identification and development of strategies that may enhance monoclonal therapeutics broadly.

Beyond Fc receptor binding, a particular antigen-specific antibody Fc glycosylation profile was linked with viral load. Specifically, subjects with low viral loads, including individuals who spontaneously control HIV to undetectable levels as well as individuals on anti-retroviral therapy, exhibited elevated levels of gp120-specific antibodies with a bisecting GlcNAc, a modification that has been actively exploited in the monoclonal therapeutics field to enhance ADCC (Umana *et al*, 1999). The addition of the bisecting GlcNAc sterically blocks the addition of core fucose, known to profoundly impact the affinity of antibodies for FcγR3a, found on NK cells (Shields *et al*, 2002). Moreover, previous studies, on a smaller group of HIV-infected individuals (Ackerman *et al*, 2013a), highlighted reduced fucose levels in spontaneous controllers and elevated bisecting GlcNAc in treated aviremic subjects, pointing to the potentially critical role for these intertwined antibody glycosylation changes on tuning antibody effector function in the context of HIV infection. Moreover, though this study was designed only to assess associations, given the intimate link between the bisecting GlcNAc and fucose on the antibody Fc domain, these data argue for a role for an enrichment of ADCC inducing antibodies in the absence of viremia. However, it is unclear whether ADCC actively contributes to reservoir control in the absence of therapy or is selectively enriched upon the resolution of virus-induced immune activation to restore viral control, but this association may point to a natural antibody Fc modification that can be selectively generated by the immune system during HIV infection to drive the rapid elimination of infected cells.

While initial biomarker discovery efforts began with the characterization of antibodies against a single antigen or pool of antigens, antibodies function as immune complexes that target distinct pathogen antigens and epitopes during infection. Here, Fc profiling was coupled to a broader integration of Fab measures, linking the remarkable biodiversity of the Fab and Fc in targeting a pathogen. Interestingly, subjects in this study who successfully controlled infection possessed antibodies targeting the structural capsid protein p24 selectively, as has been observed previously (Hogervorst *et al*, 1995). While p24 is not expressed on the surface of cells, recent data suggest that antibodies may access cytoplasmic compartments where they may induce Fc-dependent pathogen killing via cytoplasmic Fc receptors such as Trim21 (Mallery *et al*, 2010; Foss *et al*, 2015). Moreover, recent work suggests that p24-specific antibodies may be able to induce innate immune effector activities (French *et al*, 2013; Tjiam *et al*, 2015), arguing that these antibodies may not simply represent biomarkers associated with protective immunity, but may potentially also contribute to enhanced viral control. Thus, next-generation Fc-profiling efforts, that extend beyond antigen specificity and response magnitude, may point to specific antibody subpopulations in protective immunity, offering further opportunities to mine for correlates of immunity to guide vaccine and therapeutic design.

In sum, this objective analytical approach to systematically assess the spectrum of antibody biophysical features and innate immune receptor interactions, integrated within a machine-learning framework, demonstrates a novel strategy for understanding the features of functionally potent immune responses and to identify potentially mechanistic correlates of potent antibody effector function (Choi *et al*, 2015; Ackerman *et al*, 2016) and of antibody-mediated protection (Barouch *et al*, 2015; Vaccari *et al*, 2016) or

even pathology. Here, we have elucidated antibody markers of HIV viral suppression and potent effector function, pointing toward minimal biophysical signatures to support the analysis of future vaccine trials. These methods move beyond antibody titering to permit broad, yet analytically principled investigations into humoral immunity, opening a pathway toward improved mechanistic understanding of vaccination and new prospects for antibody-based prevention and therapy.

# Materials and Methods

### Study cohort

Two hundred plasma samples from adult HIV+ individuals, balanced for sex, age, and protective HLA alleles were identified from Ragon Institute HIV cohorts. The study group represented a set of four clinical classes defined by viral load and treatment. Chronic progressors, individuals who without combination anti-retroviral therapy (cART) failed to control viremia, were classified as treated (TP), or untreated (UP). The untreated group included both ART-naive subjects and those off treatment. Long-term non-progressors were classified as viremic controllers (VC), individuals with an HIV viral load between 50 and 2,000 RNA copies/ml without cART, and elite controllers (EC), a subset of viremic controllers able to maintain a plasma HIV viral load below 50 RNA copies/ml, reflecting spontaneous suppression of HIV replication to undetectable levels. Of these 200 samples, 19 were excluded for technical reasons, and 181 subject samples (45 EC, 51 VC, 44 TP, 41 UP) were included in the study. The study was reviewed and approved by the Massachusetts General Hospital Institutional Review Board; each subject gave written informed consent, and experiments conformed to principles defined in the WMA Declaration of Helsinki and the Department of Health and Human Services Belmont Report.

### Antibody purifications

IgG was enriched, and IgA, albumin, and other prevalent plasma proteins were depleted from each sample via Melon Gel purification as previously described (Ackerman *et al*, 2016). The concentration of IgG in each sample was determined by ELISA (Mabtech), and samples were diluted to a consistent total IgG concentration for testing in functional assays and in the Fc Array at constant concentrations of total IgG. Antigen-specific antibodies were purified by affinity chromatography as described previously to enable analysis of glycosylation (Brown *et al*, 2015). Briefly, Agilent Bravo Streptavidin Cartridges were loaded with approximately 50 μg of chemically biotinylated SF162 gp120 antigen and used for affinity capture. Antigen-specific antibodies from 200 μl of plasma were captured on the cartridges via centrifugation, washed, eluted with low pH (100 mM sodium citrate pH 2.9), and immediately neutralized.

### Effector function assays

The functional activity of HIV-specific antibodies was determined in a number of cell-based assays (Barouch *et al*, 2015; Ackerman *et al*, 2016; Vaccari *et al*, 2016) (Appendix Table S1). Functional assays

were performed in triplicate, with the exception of the NK activation assay, which was performed in duplicate. Replicates of assays performed using primary cells or primary plasma as a source of effectors or complement were performed across multiple donors. Averages across replicates are reported.

### ADCC

Antibody-dependent cellular cytotoxicity (ADCC) was tested using a rapid fluorescent ADCC assay, which assesses the ability of antibodies to drive primary NK cells to lyse gp120-pulsed target cells (Gomez-Roman *et al*, 2006). Briefly, SF162 gp120 protein (Immune Technology) at 60 μg/ml was used to coat CEM.NKr target cells labeled with intracellular and membrane dyes which were then co-cultured with NK cells enriched from healthy donor whole blood by negative selection (Stem Cell Technologies) effectors at an E:T ratio of 5:1. Following a 4-h incubation at 37°C and antibody at 20 μg/ml, cells were fixed, and the proportion of lysed target cells was determined by flow cytometric analysis of cells that were positive for the membrane dye but negative for the intracellular dye.

### ADNP

Antibody-dependent neutrophil phagocytosis (ADNP) was determined using an adaptation of a flow cytometry-based phagocytic assay described previously (Ackerman *et al*, 2011; McAndrew *et al*, 2011). Briefly, fluorescent, streptavidin-conjugated microspheres were coated with chemically biotinylated SF162 gp120 (Immune Technology) for 2 h at 37°C and washed with 0.1% BSA in PBS and opsonized with purified antibody for 30 min at 37°C. Neutrophils were purified from healthy donor blood anti-coagulated with acid citrate dextrose by incubation for 25 min at room temperature with equal volume of 3% dextran-500 0.9% NaCl solution, isolating the leukocyte layer and further purifying neutrophils by centrifugation over Ficoll Paque Plus (GE Healthcare) at 400 *g* for 40 min. The resulting pellet containing neutrophils was shock-treated with water to remove contaminating red blood cells. Cells were resuspended in HBSS solution without $Ca^{2+}$ or $Mg^{2+}$ and incubated with healthy donor serum diluted 1:10 in Veronal buffer saline (Fisher) + 0.1% gelatin as a source of complement and Ab-coated beads for 15 min at 37°C. Cells were then fixed with 4% paraformaldehyde solution and analyzed on a flow cytometer. A phagocytic score was derived as an integrated MFI by multiplication of the fraction of neutrophils that phagocytosed one or more opsonized beads by the MFI of this population.

### NKA

An assay to determine NK cell activation (NKA) state based on the expression of surface CD107a and intracellular production of IFNγ and MIP1β was performed as previously described. NK cells were isolated from whole blood from healthy donors using negative selection with RosetteSep (STEMCELL Technologies). Following a pulse with SF162 gp120 (60 μg/ml), T lymphoblast CEM-NKr cells and isolated primary NK cells were mixed at a ratio of 1:5, and purified IgG, anti-CD107a-phycoerythrin (PE)-Cy5 (BD), brefeldin A (10 mg/ml) (Sigma), and GolgiStop (BD) were added. After a 5-h incubation at 37°C, cells were first stained for surface markers using anti-CD16-allophycocyanin (APC)-Cy7 (BD), anti-CD56-PE-Cy7 (BD), and anti-CD3-Alexa Fluor 700 (BD) and then stained intracellularly with anti-IFNγ-APC (BD) and anti-MIP1β-PE (BD) after treatment

with Fix and Perm A and B solutions (Invitrogen). Cells were then fixed in 4% paraformaldehyde and analyzed by flow cytometry. NK cells were defined as CD3-negative and CD16-positive and/or CD56-positive, and the percent of NK cells positive for each marker was determined.

### ADCD

The ability of donor antibodies to induce complement component C3b deposition on gp120-coated target cells was assessed by flow cytometry as previously described (Barouch et al, 2015). Briefly, CEM-NKr target cells were pulsed with gp120SF162 (60 μg/ml), then incubated with purified antibody at a concentration of 100 μg/ml, and freshly harvested HIV-negative donor plasma diluted 1:10 with veronal buffer 0.1% gelatin as a source of complement for 20 min at 37°C. Following a wash with 15 mM EDTA in PBS, cells were fixed and complement deposition was detected by staining with anti-C3b-FITC (Cedarlane). The proportion of C3b-positive cells was determined based on negative controls in which heat-inactivated donor plasma was used.

### Fc array

The quantity and qualitative features of HIV-specific antibodies were evaluated using a custom multiplex array in which HIV-specific antibodies were characterized according to their titer (anti-IgG), subclass (anti-IgG 1,2,3,4), and ability to interact with innate immune antibody FcR (FcγRI, IIa, IIb, IIIa, IIIb and their major allotypic variants), initiators of the complement cascade (C1q and mannose binding lectin or MBL), and plant-derived lectin proteins, essentially as previously described (Brown et al, 2017). A diverse panel of HIV antigens coupled to carboxylated fluorescently coded magnetic beads (Luminex Corp.) was prepared essentially as previously described (Brown et al, 2012, 2017).

Biotinylated Fcγ receptors (FcγR), mannose binding lectin (MBL), and the complement cascade initiator C1q protein were tetramerized and utilized to characterize the Fc domains of each antigen-specific antibody population. FcγR and human MBL2 were produced in HEK293 cells and purified via Ni++ ion affinity and size exclusion chromatography as previously described (Boesch et al, 2014). Human C1q was purchased from Sigma (C1740). FcγRs, MBL, and C1q were minimally chemically biotinylated at a molar ratio of 5 mols biotin per mol of protein using EZ-link sulfo-NHS-SS-biotin (Pierce). The biotinylation reaction was carried out for 2 h at RT in Tris pH 8.2, with a protein concentration of 0.2 mg/ml, and free biotin was removed via buffer exchange into PBS. Lectins (Vector Laboratories) were purchased biotinylated. Immediately prior to use, each biotinylated detection reagent was mixed with a 1/4th molar ratio of streptavidin–PE (Prozyme) and diluted to a final concentration of 1.0 μg/ml in assay buffer (PBS + 0.1% BSA + 0.05% Tween). Multiplexed antibody titering and subclassing were performed as described previously (Brown et al, 2012).

A working mixture of coupled microspheres at 12.5 microspheres per bead type, per μl, were premixed in Assay Buffer. A 40 μl volume of the working microsphere mixture (500 beads of each type/well) was added to 10 μl of dilute, purified antibody sample in black, clear bottom 384-well plates (Greiner Bio One, 781906). Following incubation for 2 h at RT on an XYZ plate shaker (IKA), plates were washed five times with 65 μl of Assay Wash (1× PBS,

0.1% BSA, 0.5% Triton X-100) using a plate washing system (BioTek 405). Antigen-specific antibody was detected using the tetrameric PE-conjugated detection reagents described above, at 1.0 μg/ml, or R-phycoerythrin (PE)-conjugated anti-IgG secondary reagents (Southern Biotech), at 0.65 μg/ml, with 50 μl/well. After 1-h incubation at room temperature on a shaker, the plate was washed five times with 65 μl of sheath fluid (Luminex 40-75680), and microspheres were resuspended in 40 μl of sheath fluid.

A Bio-Plex array reader (FlexMap 3D, Bio-Plex Manager 5.0, Bio-Rad) detected the microspheres, and binding of each PE-functionalized detection reagent was measured to calculate a Median Fluorescence Intensity (MFI). MFI measurements for each antibody detection reagent—antigen pair (antibody feature), were standardized individually by subtracting the background signal, defined as the average MFI observed for each antigen microsphere set when incubated in the detector reagent in the absence of a clinical antibody sample, from each feature MFI value. These features were named according to the following convention: "detection reagent.antigen". Appendix Table S1 provides a complete list of the 41 HIV antigens tested, comprised largely of envelope variants but inclusive of a number of capsid variants and other viral proteins, and the 19 detection reagents used to interrogate the Fc domain characteristics of antigen-specific antibodies.

### Glycan analysis

A capillary electrophoresis-based technique for analyzing plasma-derived polyclonal IgG glycosylation was used to determine the relative abundance of glycan structures decorating gp120-specific and total plasma IgG, as previously described (Mahan et al, 2015). Briefly, glycans were released from IgG with the N-linked glycosidase PNGase F, were fluorescently labeled by reductive amination with 8-aminopyrene-1,3,6-trisulfonic acid, separated from unreacted dye, and analyzed on an Agilent 3130XL ABI DNA sequencer. Peak identities were determined by exoglycosidase reactions and use of glycan standards, and the prevalence of each glycan species was determined by peak area integration with a custom script. The prevalence of 19 individual glycoforms and six summary glycosylation state assessments were determined (Appendix Table S1). Summary glycosylation characteristics, such as level of galactosylation (G) and sialylation (S), and presence of absence of fucose (F) or bisecting GlcNAc (B), were compiled by summing individual glycoforms. Unadjusted, two-tailed Pearson correlation coefficients, assuming Gaussian distributions, and unadjusted, two-tailed t-tests assuming equal variances between groups, are reported.

### Data analysis and visualization

Input data are compiled in Dataset EV1 in four separate spreadsheets (comma-separate values, csv, format): functional measurements, glycan measurements, Luminex measurements for IgG detection reagents ("titer"), and Luminex measurements for other detection reagents ("Fc Array"). Each spreadsheet has a single row per subject with a single column per feature.

Basic data analysis and visualization were performed using GraphPad Prism along with in-house scripts developed for the R statistical computing environment (2013) supported by standard R packages caret, ggplot2, and gplots.

Graphical networks were generated using Cytoscape (2013; Shannon *et al*, 2003), with nodes and edges programmatically generated by an R script so as to include for each subject group just those pairs of features having at least a weak correlation over the group's subjects, namely an unadjusted Pearson correlation coefficient (PCC) magnitude of 0.4 at an unadjusted *P*-value of 0.05, and assuming Gaussian distributions. Edge colors were linearly mapped from −1 (orange) to 0 (white) to 1 (blue) and line styles set according to three different *P*-value thresholds (< 0.05: thin dotted; < 0.01: medium dashed; 0.001: thick solid). Cytoscape files are provided in Dataset EV2.

## Predictive modeling

While previous studies have shown that a number of different algorithms can be effectively used to model antibody activities (Choi *et al*, 2015), for the sake of model simplicity and interpretability we employed here generalized linear models (GLMs). GLMs encompass statistical learning methods that model a response, here subject classes and functional activities, in terms of weighted sums of feature values, here Fc Array and glycan measurements, in a manner generalizing beyond the assumption of normally distributed error as with standard linear models (Nelder & Weddebum, 1972). Such models can be productively complemented with regularization approaches designed to reduce variance and redundancy and mitigate the potential for overfitting "wide" data (Hastie *et al*, 2009). In this study, we performed single response linear regression to predict functional assay data, binary logistic regression to classify subject group pairs (viremic versus aviremic, controller versus progressor), and multinomial logistic regression to classify all four subject groups (EC, VC, TP, UP).

### Preprocessing
Prior to modeling, each antibody feature was independently scaled and centered to a mean of 0 and a standard deviation of 1 as previously described (Choi *et al*, 2015) to facilitate interpretability of model coefficients.

### Modeling
All models were trained and evaluated using the glmnet R package (Friedman *et al*, 2010) for regularized generalized linear modeling. Glmnet systematically evaluates the effect of a regularization parameter $\lambda$ on prediction accuracy (here misclassification error or mean-squared regression error), in a cross-validation setting (here 10-fold). Glmnet further allows varying an elastic net parameter $\alpha$ to trade off between reducing the number of included features (Lasso *L1* penalty) and managing collinearity (ridge *L2* penalty), allowing a balance between the two (elastic net *L1,L2* penalty) (Friedman *et al*, 2010). The elastic net approach is particularly well suited for high-dimensional datasets, where the number of features/predictors is much greater than the number of observations and many features may be highly correlated (Hastie *et al*, 2009; Friedman *et al*, 2010), as is the case in this study. In order to obtain smaller feature sets driving the models, $\alpha = 0.8$ (more Lasso-like) was used for the presented results; similar performance and feature sets were obtained with $\alpha = 0.4$ (more ridge-like) and 1.0 (Lasso), suggesting good generalization.

### Model training, evaluation, and selection
The glmnet modeling process was repeated 100 times, for each replicate training models and evaluating their 10-fold cross-validated prediction error over a range of $\lambda$ values. Glmnet's assessment of the impact of $\lambda$ enables identification of two natural choices for the parameter: $\lambda_{min}$, the value that achieved the minimum cross-validated error, and $\lambda_{1se}$, the largest value of $\lambda$ yielding error within one standard error of the minimum (Friedman *et al*, 2010). In general, we found the performance using these two settings to be quite similar and the features selected at $\lambda_{1se}$ to be a subset of those selected at $\lambda_{min}$. For consistency, reported prediction accuracies were based on $\lambda_{min}$. In order to enable discussion of details of a specific modeling run and a specific model, a "representative replicate" for each task was selected as that obtaining median accuracy over 100 replicates, and its "representative model" was selected as the full glmnet model (i.e., trained on all data) at the $\lambda_{min}$ for that replicate.

### Overall evaluation and robustness
The 100 repetitions enabled assessment of the robustness of the modeling approach, in terms of variance in prediction accuracy. To further evaluate reliability, in terms of how well one could expect a model to perform at random, we performed a permutation test (Ernst, 2004), producing 100 cross-validated models learned using the same input data and elastic net parameter, but with the response variable randomly shuffled.

### Modeling code
A single R script encapsulating all modeling steps is available in Code EV1. The script takes the experimental data files, along with meta information (antigen, detection reagent, and glycan grouping information, subject classes, and coloring schemes) collected in Dataset EV1, and produces a set of output files that are hierarchically organized by the $\alpha$ value for glmnet, the input data type (e.g., Luminex, glycan, etc.), the output data type (e.g., ADCC, four-class subject classification, etc.), and the choice for the glmnet $\lambda$ parameter ($\lambda_{min}$ or $\lambda_{max}$). The output files include the coefficient path plot; a summary of the overall performance for input data and permuted input data; and both pdf-format plots and raw csv files for the cross-validated performance over varying $\lambda$ values, predicted versus observed values for the representative replicate, and coefficients for the full model.

### Modeling results
Complete output files for models presented ($\alpha = 0.8$, $\lambda_{min}$) are supplied in Dataset EV3.

## Data availability

Input data are compiled in Dataset EV1, and cytoscape files are available in Dataset EV2. Model outputs are provided in Dataset EV3, and modeling scripts are provided in Code EV1.

**Expanded View** for this article is available online.

## Acknowledgements
These studies were supported in part by the Bill and Melinda Gates Foundation (OPP1032817) and the National Institutes of Health (R01AI102691 and P01AI120756). We dedicate this article to the memory of our friend and respected colleague, Dr. Chris Scanlon, for his valuable contributions to the genesis of this project.

## Author contributions

EPB, TJS, AM, and AEM collected and analyzed data. GA, FN, and MEA oversaw experiments. BDW contributed samples. KGD and CBK conducted data modeling. GA, KGD, CB-K, and MEA wrote the paper. All authors provided critical review.

## Conflict of interest

The authors declare that they have no conflict of interest.

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
