## [Review Process File · Molecular Systems Biology]

High-resolution definition of humoral immune response correlates of effective immunity against HIV

Galit Alter, Karen G. Dowell, Eric P. Brown, Todd J. Suscovich, Anastassia Mikhailova, Alison E. Mahan, Bruce D. Walker, Falk Nimmerjahn, Chris Bailey-Kellogg and Margaret E. Ackerman

Review timeline:

Submission date:	18 July 2017
Editorial Decision:	13 October 2017
Revision received:	9 January 2018
Editorial Decision:	12 February 2018
Revision received:	26 February 2018
Accepted:	27 February 2018

Editor: Maria Polychronidou

Transaction Report:

1st Editorial Decision

13 October 2017

Thank you again for submitting your work to Molecular Systems Biology. As you will see below, both reviewers appreciate that the presented findings seem interesting. They raise however a series of concerns, which we would ask you to address in a revision.

The reviewers' recommendations are rather clear so I think that there is no need to repeat the points listed below. Please do not hesitate to contact me in case you would like to discuss/clarify any of the points listed by the referees.

REVIEWER REPORTS

Reviewer #3:

Alter et al. applied their previously-described "systems serology" antibody biophysical property profiling approach to a large cohort of HIV-infected individuals. The resulting high-dimensional data was used to build models that differentiate between the humoral immune responses of elite controllers, viremic controllers, and treated and untreated disease progressors. The analysis yielded a number of intriguing differences between subject groups provide rich datasets for further analyses. The data/models did an impressive job in discriminating between study groups, as well as in predicting antibody effector functions.

A more thorough explanation of analyses and figures are needed in many areas. Many of the figures were not explained in adequate detail for scientists without systems biology/statistics expertise. For

example, Figure 2B-D were not detailed outside of the figure legend (what are confusion matrices or LOD scores?).

The manuscript would also be improved by more clearly highlighting the predictive variables in the models, providing further biological interpretation, and bridging these with some of the main biological findings.

The authors noted that aviremic individuals had an enrichment of env-specific antibodies with bisected glycans compared to viremic individuals, while this was not the case with the total antibody response. The authors tout this as an example of the differences that are revealed by biophysical modeling of the study groups. However, this "example" is not readily apparent in any of the model datasets (or at least not pointed out in sufficient detail). I assume that the bottom panel of Figure S2 is in fact explaining this viremic/aviremic model (please also be consistent with group names), but S2 is never mentioned in the manuscript text. Further, no glycan variables appear to contribute to the model (Figure S2C?). In Figure 3, they present this observation using P-values - which elicits concerns that they have not adequately performing a multiple-hypothesis testing correction given the large number of variables that they are testing. Further, the authors suggest relevance of these env-specific bisected-glycan antibodies by citing other studies that have shown such antibodies exhibit enhanced FCgammaR3 recognition and ADCC activity. Thus, are the env-specific bisected-glycan antibodies correlated with ADCC activity and FCgammaR3 ligation propensity data collected in this study? Are these measures different between the viremic and aviremic groups?

Similarly, another interesting result is that ECs have elevated levels of p24 antibodies, which is readily apparent in the volcano plot. However, is this true in the classification model? I cannot tell, as the legend to Figure S2C (controller/progressor) is not legible.

These two biological observations, while interesting, do not appear to be "Examples of differentiating humoral responses features identified by classification models" as the Figure 3 title states. Rather they are examples of observations possible with the high-dimensional data, which confirm or support previous observations / hypotheses. That is fine and still important - however, the authors should reframe these findings accordingly, or better highlight how the models point to these observations.

A number of figure axes labels are illegible (Figure S2C, S3D). The figure 6 legend should note that the plots are shown with labeled variables in Figure S3.

As a more general point, the manuscript should do a better job of emphasizing that the tested variables are CORRELATED with viral control / titer, but are not necessarily causal. For instance, it is possible that the immune signatures reflect reduced viremia rather than cause it.

Finally (and crucially), the authors do not provide adequate data and analysis code. This paper is ultimately a computational analysis of large datasets, but neither the datasets or the computer code are available at least as far as I can tell from the manuscript. All of the underlying datasets and computer code should be made available as supplementary material in a usable format: not PDF images, but CSV / text files and computer code files. If this cannot be accommodated in the journal supplementary format rules, then they should be posted on DataDryad or GitHub or some similar repository designed for hosting code and data.

Reviewer #4:

This is an interesting paper that presents new high-dimensional humoral immunoprofiling approaches and that applies those to elucidate hallmarks of effective HIV-1 viral control. The authors use a combination of new experimental and computational approaches to integrate humoral data on functional, effector function, complement fixation and phagocytosis. Overall, I find the methods and results interesting. This being said, I do have a few questions/concerns that I present below:

Major comments:

Figure 4: I wonder how robust the networks/findings are in this figure? What happens if you were to modify the $|r| > 0.4$ and $p < 0.05$ thresholds? Would the findings be consistent? It would be good to perform a quick sensitivity analysis. Perhaps, use different values of r to see how it changes. Also, how many of the p -values would survive multiple testing. I understand that this is mostly visual and exploratory, but it would be good to understand how strong these interactions are. Perhaps this could be discussed to highlight the fact that many of those are likely false positives.

Redundancy across assays: As shown on Figure 5, it is clear that some of the assays/variables are highly correlated. I was wondering if the prediction accuracy presented in Figure 2 could be improved if one were to downselect the variables to reduce correlation. While the Elastic Net can deal with collinearity, its performance can be greatly affected by it. Also, scientifically speaking it would be great to come up with a minimal set of variables/assays that capture the breadth of antibody function while minimizing redundancy across variables. I understand that this is difficult to do based on a single dataset, but the authors might have some other data that could be used for that. I would at least discuss it.

Figure 5: As discussed above, this is an interesting figure/result. I would propose to use the mean squared error (MSE) for the y -axis as it is more standard for evaluating prediction accuracy. Related, for 5b, report r^2 instead of r .

Use of biological replicates in Figure 5. I don't really understand why the correlation between biological replicates would be an upper bound. Are these technical or biological replicates? If biological, I would argue that the correlation should probably be lower as there might be substantial variability across subjects. I think this needs to be clarified.

Minor comments:

Figure 4: The legend in this figure should say $|r| > 0.4$ and not $r > |.4|$.

Code and data availability: Given the importance of the computational analyses performed, it would be great to share the data and code for full reproducibility.

We thank the reviewers for their thoughtful comments, questions, and suggestions. Responses to each point made in review are described below, and modifications made the manuscript have been tracked in the resubmission. We believe that the manuscript has been substantially improved based on addressing these comments, and we further note that the revised supplemental materials now includes raw data, code, and model outputs.

Reviewer #3:

Alter et al. applied their previously-described "systems serology" antibody biophysical property profiling approach to a large cohort of HIV-infected individuals. The resulting high-dimensional data was used to build models that differentiate between the humoral immune responses of elite controllers, viremic controllers, and treated and untreated disease progressors. The analysis yielded a number of intriguing differences between subject groups provide rich datasets for further analyses. The data/models did an impressive job in discriminating between study groups, as well as in predicting antibody effector functions.

A more thorough explanation of analyses and figures are needed in many areas. Many of the figures were not explained in adequate detail for scientists without systems biology/statistics expertise. For example, Figure 2B-D were not detailed outside of the figure legend (what are confusion matrices or LOD scores?).

In the revised manuscript we have taken care to better introduce terms associated with our analysis approach that may be unfamiliar to readers, as well as to discuss the figures in greater depth.

See for example, page 4-5: "Given the rich data collected, a machine learning approach was employed to identify combinations of a humoral response features able to predict subject class and antibody effector function. Cross-validated classifiers trained to distinguish subject groups identified minimal sets of antibody features that accurately and robustly discriminated among all four subject groups, between viremic and aviremic subjects, and between controllers and progressors (**Fig.2**). Subject class was determined by a class score (LOD, or Log₂ Odds Ratio) that defined the relative likelihood of a given subject's assignment to one as compared to other classes. Differentiation across EC, VC, TP and UP groups was accomplished with approximately 60% accuracy, as compared to the approximately 25% accuracy expected at random based on class size for this 4-group differentiation model, and as observed when scrambled study data was used as a modeling input (**Fig.2a**). Approximately 75% accuracy was observed for the 2-way classifications aimed at differentiation of subjects by progression (EC and VC versus TP and UP) or by viremia (EC and TP versus VC and UP), as compared to the approximately 50% accuracy expected by chance, or again, observed when study data was permuted prior to learning (**Fig.2a**). Confusion matrices (**Fig.2b-d**), which compare actual and predicted classifications, indicated that with the exception of UPs, who could not be confidently differentiated from TPs or VCs, most classes were predicted well. While there was little evidence of systematic confusion, among misclassified VCs and TPs, most were modeled to be UPs, consistent with the difficulty noted in classification of this group, and suggestive that they may have a less distinct humoral profile as a group. Further, class score can be considered a measure of model confidence (**Fig.2b-d**). Whereas in the 2-way classifications misclassified subjects often had marginal class scores, in the 4-group differentiation models, incorrect predictions were often made confidently. For example, misclassified ECs tended to be confidently predicted as either TPs or VCs, suggestive of the existence of some subjects with profiles that are considerably more consistent with these classes."

The manuscript would also be improved by more clearly highlighting the predictive variables in the models, providing further biological interpretation, and bridging these with some of the main biological findings.

Based on this thoughtful suggestion we have expanded the manuscript text to more fully describe and interpret the features identified as predictive of group and functional activity.

See for example, page 6: "For example, numerous bisected glycoforms contributed to the differentiation of aviremic from viremic subjects when the glycosylation profiles of HIV envelope-specific antibodies were used to predict viral load status (**Appendix Fig S3**), suggesting a difference in the level of

antibody bisection between groups.... Whether this glycosylation state difference associated with viral load may be a cause or effect of viremic status remains to be determined, but potential mechanistic relevance to antiviral humoral immunity is suggested by previous studies demonstrating that antibodies with bisected glycans exhibit enhanced FcγR3 recognition and ADCC activity (31), a mechanistic link that was further explored within this data set in models of antibody function. ”

Page 6: ”Analysis of the Fab-specificity of features contributing to classification of ECs but not observed in modeling viremia status (**Appendix Figures S1, S2, and Dataset EV2**), pointed toward recognition of the virus capsid (p24/Gag) as a differential marker between subjects with immune-mediated as opposed to pharmacologic viral suppression.”

The authors noted that aviremic individuals had an enrichment of env-specific antibodies with bisected glycans compared to viremic individuals, while this was not the case with the total antibody response. The authors tout this as an example of the differences that are revealed by biophysical modeling of the study groups. However, this "example" is not readily apparent in any of the model datasets (or at least not pointed out in sufficient detail).

We apologize for the lack of clarity regarding how the individual examples selected for inclusion in Figure 3 relate to functional and class models. The manuscript text and figure set has been significantly expanded to specifically highlight the features that are uniquely selected by the model and make the linkage more clear, as indicated by the example above, and in greater detail below.

I assume that the bottom panel of Figure S2 is in fact explaining this viremic/aviremic model (please also be consistent with group names), but S2 is never mentioned in the manuscript text. Further, no glycan variables appear to be contribute to the model (Figure S2C?). In Figure 3, they present this observation using P-values - which elicits concerns that they have not adequately performing a multiple-hypothesis testing correction given the large number of variables that they are testing. Further, the authors suggest relevance of these env-specific bisected-glycan antibodies by citing other studies that have shown such antibodies exhibit enhanced FCγR3 recognition and ADCC activity. Thus, are the env- specific bisected-glycan antibodies correlated with ADCC activity and FCγR3 ligation propensity data collected in this study? Are these measures different between the viremic and aviremic groups?

A number of issues are raised here, which have been addressed as follows:

- In the revised manuscript we have made more explicit reference to Appendix Figure S2 (which relates to the classification results presented in Figure 2b). This figure had been poorly referenced as “Figs.S1,2” in the original submission.
- We have ensured that groups are referred to consistently throughout the manuscript, figures, and Appendix (ie: replacing “nonviremic” with “aviremic”).
- We have provided detail on the basis for inclusion of the specific examples selected for Figure 3 and provided the “missing link” the reviewer is looking for in terms of relating these individual measures to the models as follows:
 - For Figure 3a: new text in the results and a new supplemental figure (Appendix Figure S3) describing the use of glycan data to predict aviremic vs. viremic classes that raises the hypothesis that glycan bisection may differ among groups is included to provide the basis for investigating this parameter in the context of these groups.
 - For Figure 3b: an expanded description of the linkage between the model of subject class whose in response to the more direct reviewer comment on this matter below.
- Given the basis for the plots in Figure 3 is now (better) provided, we have not addressed the multiple hypothesis testing concern raised as we hope it is now clear that these features were not selected on the basis of their p-values, but were investigated based on modeling results.
- Beyond the literature supporting the biological relevance of a difference in antibody bisection, the importance of this difference to the sample set evaluated within this study is now supported by drawing attention to:
 - Figure 6d, which reports the contribution of IgG bisection to the model of ADCC activity (see also Appendix Figure S6), and the correlation of bisected glycoforms with ADCC

- activity (see also Appendix Figure S7).
- Figure 6d, which also illustrates the predominance of Fcγ3R ligation propensity assessments to contribute to ADCC activity models.
- The correlation of Fcγ3R ligation with ADCC, as compared to the correlation between IgG subclass levels or C1q ligation and ADCC is illustrated for the purpose of review, below in **Response to Review Figure 1**.
- Appendix Figure S2, in which the HIV gp120-specific positive coefficient with greatest weight in viremic/aviremic classification is an Fcγ3R measurement.

Response to Review Figure 1: Correlation heatmap depicting relative relationships between ADCC activity and Fc Array measurements among ECs.

Collectively, these relationships support both the relevance of bisection to ADCC activity, and viremic/aviremic group differentiation, within this study, and provide the basis for the inclusion of this feature in Figure 3, and the justification for the description in the figure legend as an example of “differentiating humoral response features identified by classification”.

Similarly, another interesting result is that ECs have elevated levels of p24 antibodies, which is readily apparent in the volcano plot. However, is this true in the classification model? I cannot tell, as the legend to Figure S2C (controller/progressor) is not legible.

The elevation of p24-specific antibodies among ECs evident in the volcano plot is indeed reflected in the models of subject class. In the 4 way classification models, of the 16 features with positive coefficient weights for the EC group, 9 represent p24/Gag-specific antibody measurements. In fact, 2 of the top 3 positive coefficients are p24/Gag-specific.

We apologize for the lack of label clarity. Though the supplemental pdfs supplied can generally be clearly read under magnification, we recognize that the small print required to fit the labels is a limitation to the utility of the supplemental graphs when printed. Given the difficulty in reading labels on supplemental figures at print size, and the potential interest in and utility of the data underlying these graphs, we have now included tables reporting the features and coefficient weights for models in the supplemental data.

Relevant to the question raised by the reviewer, we have used these tables to generate a list (**Response to Review Table 1**) of the feature coefficients for the EC group in order of decreasing magnitude, with p24/Gag-specific antibody features indicated in yellow.

	EC
IgG1.IIIb.pr55.Gag	0.379
FcγRIIIa.Chiang.Mai.gp120	0.379
C1q.p24.HXBc2	0.319
FcγRIIb.HIV1.Vif	0.306
IgG3.HIV1.Integrase	0.256
FcγRIIIa.R131.p24.HXBc2	0.241
FcγRIIIb.SH.p24.HXBc2	0.170
SNA.HIV1.Nef	0.167
FcγRIIIb.SH.IIIb.pr55.Gag	0.130

FcgRIIIb.IIIb.pr55.Gag	0.116
FcgRIIIb.SH.HIV1.p7	0.113
IgG4.HIV1.Rev	0.086
IgG3.93TH975.gp120	0.078
C1q.p24.IIIb	0.068
LCA.p24.HXBc2	0.040
FcgRIIIa.F158.IIIb.pr55.Gag	0.025
IgG1.gp120.SF162	-0.001
FcgRIIIa.V158.gp140.UG21	-0.015
VVL.gp140.CN54	-0.027
C1q.IIIb.pr55.Gag	-0.041
VVL.HIV1.Nef	-0.053
PNA.gp140.CN54	-0.076
MBL.HIV1.Vif	-0.111
SNA.gp41.HXBc2	-0.115
C1q.gp120.RSC3	-0.139
IgG1.gp140.CN54	-0.175
IgG3.HIV1.Nef	-0.188
MBL.93TH975.gp120	-0.188
C1q.gp120.IIIb	-0.194
LCA.gp41.HXBc2	-0.204
VVL.HIV1.Rev	-0.257
FcgRIIIa.gp120.SF162	-0.310
VVL.p24.HXBc2	-0.336
C1q.gp120.ZM109F	-0.338
FcgRIIIa.SOSIP	-0.930

Response to Review Table 1: Feature coefficients for classification of the EC group in 4-way classification models. Antibodies specific for p24/Gag are indicated with yellow shading. Line indicates the boundary between features with positive and negative coefficients. (Related to Figures 2 and S1.)

These two biological observations, while interesting, do not appear to be "Examples of differentiating humoral responses features identified by classification models" as the Figure 3 title states. Rather they are examples of observations possible with the high-dimensional data, which confirm or support previous observations / hypotheses. That is fine and still important - however, the authors should reframe these findings accordingly, or better highlight how the models point to these observations.

We hope that the expanded description of how these selected differences relate to classification models, as described above, provides suitable justification for the Figure 3 title.

A number of figure axes labels are illegible (Figure S2C, S3D). The figure 6 legend should note that the plots are shown with labeled variables in Figure S3.

We apologize for the label sizes on supplemental figures at print size. To address this and related comments, and given potential interest in and utility of the data underlying these graphs, we have now included tables reporting the feature labels and coefficient weights for all models in the supplemental data.

We have also now added a note to the Figure 6 legend that Appendix Figure S6 (as renumbered from S3 in revision), as well as the model output tables, report the full text labels for the features used in these models.

As a more general point, the manuscript should do a better job of emphasizing that the tested variables are CORRELATED with viral control / titer, but are not necessarily causal. For instance, it is possible that the immune signatures reflect reduced viremia rather than cause it.

In the revised manuscript, we have emphasized that our findings are associative and that causality cannot be determined from this study.

See for example:

Page 4: "...suggesting the value of comprehensive IgG Fab, Fc, and functional profiling to inform the identification of humoral mechanisms of action and **associative** relationships to viral suppression."

Page 4: "By modeling their interactions to define the underlying principles by which antibodies collaborate and/or compete to direct the overall antiviral activity of the humoral response, we define new biomarkers **associated with** antiviral antibody effector function, aviremia, and non-progression."

Page 6: "Whether this glycosylation state difference **associated with viral load** may be a cause or effect of viremic status remains to be determined, ..."

Page 10: "Thus, **while observations using this approach are necessarily only associative**, an integrated Fc-functional and deep biophysical profiling offers a unique opportunity for the identification and development of strategies that may enhance monoclonal therapeutics broadly."

Page 10: "Moreover, **though this study was designed only to assess associations**, ..."

Finally (and crucially), the authors do not provide adequate data and analysis code. This paper is ultimately a computational analysis of large datasets, but neither the datasets or the computer code are available at least as far as I can tell from the manuscript. All of the underlying datasets and computer code should be made available as supplementary material in a usable format: not PDF images, but CSV / text files and computer code files. If this cannot be accommodated in the journal supplementary format rules, then they should be posted on DataDryad or GitHub or some similar repository designed for hosting code and data.

As requested, the revised manuscript includes supplemental files that include 1) the raw experimental data (dataset EV1), 2) model outputs (dataset EV2), and 3) the code used to generate the results reported (modeling scripts EV3).

Reviewer #4:

This is an interesting paper that presents new high-dimensional humoral immunoprofiling approaches and that applies those to elucidate hallmarks of effective HIV-1 viral control. The authors use a combination of new experimental and computational approaches to integrate humoral data on functional, effector function, complement fixation and phagocytosis. Overall, I find the methods and results interesting. This being said, I do have a few questions/concerns that I present below:

Major comments:

Figure 4: I wonder how robust the networks/findings are in this figure? What happens if you were to modify the $|r| > .4$ and $p < 0.05$ thresholds? Would the findings be consistent? It would be good to perform a quick sensitivity analysis. Perhaps, use different values of r to see how it changes. Also, how many of the p -values would survive multiple testing. I understand that this is mostly visual and exploratory, but it would be good to understand how strong these interactions are. Perhaps this could be discussed to highlight the fact that many of those are likely false positives.

We agree that such networks can be a useful visualization but can represent only a snapshot of the relationships between parameters. We provide below some sensitivity analysis below, have added

more significance information to the figure, and have amended the text to better call attention to the limitations inherent to use of this type of analysis that the reviewer has noted as follows:

See page 7: “Balancing the competing desires of discovery and confidence, plots depicting relationships between glycoforms and Fc Array measurements with correlation coefficients exceeding an absolute value magnitude of 0.4 and an uncorrected p-value of 0.01, representing the top 7% of correlations in strength and significance, were generated.”

Additionally, we note the correlation strength is indicated by color scale in the figure. While values with $|r| < 0.4$ are excluded from the visualization, we consider this to be a reasonable threshold based on balancing discovery and confidence. This choice results in presentation of feature:feature relationships in the top ~7% across the data set (**Response to Review Figure 2**).

Response to Review Figure 2: PCC magnitude and significance histograms for all feature:feature relationships, as compared to those selected for inclusion in Figure 4’s network plots.

Additionally, we have revised Figure 4 such that line style now indicates the significance of the correlation, with thin dotted lines indicating $p < 0.05$, thick dashed lines indicating $p < 0.01$, and thick solid lines indicating $p < 0.001$ (**Response to Review Figure 3**).

Response to Review Figure 3: Zoomed in view of network plots showing depiction of correlation strength and significance.

Redundancy across assays: As shown on Figure 5, it is clear that some of the assays/variables are highly correlated. I was wondering if the prediction accuracy presented in Figure 2 could be improved if one were to downselect the variables to reduce correlation. While the Elastic Net can deal with collinearity, its performance can be greatly affected by it. Also, scientifically speaking it would be great to come up with a minimal set of variables/assays that capture the breadth of antibody function while minimizing redundancy across variables. I understand that this is difficult to do based on a single dataset, but the authors might have some other data that could be used for that. I would at least discuss it.

We agree that many of the antibody characteristic variables are highly correlated (as is expected given that most antibody types can interact well with most Fc receptors), and that correlated variables can adversely impact model performance. In a previous study we reported an exploration of several different dimensionality reduction approaches, and several different modeling approaches, and found that neither had a strong impact on models of antibody activity performance-wise (Choi et al, PLoS Comp Bio 2015 - <http://journals.plos.org/ploscompbiol/article?id=10.1371/journal.pcbi.1004185>).

Here, to investigate the sensitivity of the Elastic Net approach to collinearity among features, we evaluated alpha setting of 1.0 and 0.4, ranging from lasso-like to ridge-like, respectively, and flanking the setting of alpha = 0.8 selected for models presented in the manuscript. We find that model performance was not highly sensitive to this parameter (**Response to Review Figure 4**).

Response to Review Figure 4: Scatterplot of model accuracy outputs for antibody functions (left) and classification models (right, includes 2-way and 4-way classifications). Performance with actual data is shown in black, performance with permuted data is shown in green.

Beyond these technical considerations, as the reviewer suggests, development of a minimal (and non-redundant) feature set that could encapsulate the breadth of antibody function is of high interest given the multiple antibody functions that have correlated with vaccine-mediated protection in non-human primate models across multiple vaccine approaches and regimens. As the reviewer suggests, we agree that analysis of multiple data sets, particularly those from vaccine studies (as opposed to the natural infection cohort evaluated here) would be important in order to have high confidence in the value of a reduced feature set for this purpose. Our study relates only these HIV infected subject populations, including rare controllers, highlighting key features of antibodies that may enhance their activity; we now mention that future studies aimed at further exploring the unique antibody features/profiles in in other cohorts would be of high interest.

See page 9: “However, while novel platform technologies, such as the recently described Systems Serology approach (Ackerman et al, 2017) can provide a comprehensive glimpse at the broad view of the landscape of functional responses of pathogen-specific antibodies, approaches to specifically define minimal physical biomarkers, **which can be tailored to maximize information and reduce experimental redundancy**, which are more easily interrogated in validated assays, and that are associated with desirable clinical outcomes, **need to be further implemented to better define cross-cohort, cross-regimen, and cross-pathogen principles of protective humoral immunity**.”

Figure 5: As discussed above, this is an interesting figure/result. I would propose to use the mean squared error (MSE) for the y-axis as it is more standard for evaluating prediction accuracy. Related, for 5b, report r^2 instead of r .

We agree that MSE is a standard metric for prediction accuracy. However, we sought to explicitly compare the modeling results with experimental replicates, for which correlation coefficients are more typically reported. We also desired to enable comparisons between functional assays, whereas interpretation of MSEs across functions is complicated by the different activity ranges and magnitudes for each assay. Given this intention, we have retained the original y-axis label in Figure 5, and note that MSE values (\pm SD over cross-validation replicates and folds) are provided for each model in Figures S1, S2, S3, and S4. Because these figures include MSE over varying values of lambda, they also provide useful information regarding interpretation of the MSE magnitudes.

As requested, r^2 values are now reported (Figures 5b, S3, and S4).

Use of biological replicates in Figure 5. I don't really understand why the correlation between biological replicates would be an upper bound. Are these technical or biological replicates? If biological, I would argue that the correlation should probably be lower as there might be substantial variability across subjects. I think this needs to be clarified.

We apologize for the confusion regarding our description. Consistent with the figure labeling, correlations between biological replicates are shown. Given the confusion this comparison and its description raised, we have tempered our description/statement as follows:

See page 7: "Prediction accuracy, the correlation between modeled and observed activity, was generally as good as the degree of correlation observed between assay replicates, which **provides a reasonable benchmark for how well a model might be expected to perform.**"

Minor comments:

Figure 4: The legend in this figure should say $|r|>0.4$ and not $r>|.4|$.

We apologize for this embarrassing error. It has been corrected.

Code and data availability: Given the importance of the computational analyses performed, it would be great to share the data and code for full reproducibility.

As requested, the revised manuscript includes supplemental files that include 1) the raw experimental data (dataset EV1), 2) model outputs (dataset EV2), and 3) the code used to generate the results reported (modeling scripts EV3).

Thank you for sending us your revised manuscript. We have now heard back from the two reviewers who were asked to evaluate your study. As you will see below, they are both satisfied with the modifications made and think that the study is now suitable for publication.

Before we formally accept your manuscript for publication, we would like to ask you to address the editorial issues below.

REVIEWER REPORTS

Reviewer #4:

I am happy with all changes to the paper. As far as I am concerned, it can be published as is.

Reviewer #3:

The authors meticulously addressed all of our concerns, and did a commendable job providing the raw data, analysis code, and output figures.

Corresponding Author Name: Ackerman, Margaret E

Manuscript Number: MSB-17-7881